# How frailty index impacts death in chronic kidney disease: A retrospective observational investigation

Mengmei Xiong, Xiaoyan Lu*

Department of Nephrology, Hangzhou TCM Hospital Affiliated to Zhejiang Chinese Medical University, Hangzhou, China

* lxy13858056064@163.com

## Abstract

### Background

The present study assessed the link between the frailty index and deaths from all causes or specific causes in patients with chronic kidney disease (CKD). The study data were derived from the National Health and Nutrition Examination Survey (NHANES) (1999–2018) involving 3262 CKD patients.

### Methods

We used 53 multifaceted assessment instruments to measure frailty degree. Multivariable Cox regression analysis was performed, and hazard ratio (HR) and 95% confidence interval (CI) were calculated.

### Results

The median frailty index was 0.166 (interquartile range [IQR]: 0.01 to 0.665). During the median follow-up period of 9.4 years, a total of 1102 deaths from all causes were recorded, which included 196 cancer-related deaths and 402 heart disease-related deaths. Patients in the highest frailty index tertile showed more risk of dying from cardiovascular illness (adjusted HR 2.00, 95% CI 1.52–2.64), all causes (adjusted HR 1.81, 95% CI 1.54–2.13), and cancer (adjusted HR 1.59, 95% CI 1.09–2.33). The increase in cardiovascular-related deaths, all-cause-related deaths, and cancer-related deaths was 112% (P < 0.001), 85% (P < 0.001), and 72% (P < 0.001), respectively, for every logarithmic unit increase in the frailty index. These associations remained robust in stratified tests by sex, body mass index, age, race, diabetes history, hypertension status, and CKD stages.

**Data availability statement:** All original data could be publicly available at the NHANES database: https://wwwn.cdc.gov/nchs/nhanes/default.aspx. The data supporting the conclusions of this article are provided in the Supporting Information file named "S1 Data".

**Funding:** This research was supported by the Construction Fund of Key Medical Disciplines of Hangzhou under Grant No. 2025HZGF12, 2025HZPY05. The funder had no role in the study design, data collection and analysis, decision to publish, or preparation of the manuscript.

**Competing interests:** The authors have declared that no competing interests exist.

## Conclusions

Our findings suggest that among patients with CKD, the frailty index is linked to both cause-specific and all-cause deaths. A key element of managing CKD should be frailty intervention as the frailty index may indicate prognosis in these patients.

## Introduction

Chronic kidney disease (CKD) is an irreversible and ongoing clinical syndrome, marked by the kidneys' inability to filter waste from metabolism and eliminate extra fluid [1]. By 2040, CKD is expected to rank as the fifth leading cause of death worldwide, with 10% of adults having some CKD type and causing 1.2 million deaths annually [2]. The clinical burden of CKD is not limited to end-stage renal disease (ESRD), but it is also one of the primary causes of accelerated cardiovascular complications and high mortality [3]. Therefore, CKD induces substantial healthcare expenditure and poses a huge socioeconomic burden [4], making it a serious public health issue [5].

Frailty is a condition arising from deficiencies across various interconnected physiological systems. It impairs a person's capability to respond to or recuperate from physiological stressors, indicating disrupted homeostatic ability [6]. In the elderly, frailty is linked to a greater risk of falls, cognitive decline, hospitalization, and mortality as well as having various other adverse health-related consequences [7]. Frailty not only affects older individuals, but it also appears in middle-aged people having chronic illnesses or in those who experience functional limitations. Frailty is generally evaluated by extensive assessment of multiple health parameters, including cognitive ability, capacity to engage in activities of daily living (ADL), physical attributes, comorbidities, and test outcomes. The frailty index serves as a comprehensive instrument for evaluating health status, providing insights into an individual's general well-being and vulnerability to various health conditions [8]. In the past few years, the notion of frailty has garnered significant attention as a standalone prognostic factor within the realm of nephrology.

A study involving 886 patients with CKD with varying levels of kidney function showed that as kidney function declines, the extent of frailty worsens, with a corresponding increase in patient death [9]. Frailty is a very common condition in CKD patients, and it is predictive of negative outcomes such as all-cause- and cardiovascular-related deaths as well as dialysis [10]. Previous research has shown that the frailty status of patients with CKD is closely associated with the risk of various cancers [11]; moreover, frailty increases the risk of mortality in cancer survivors [12,13]. However, there is a scarcity of research on how the frailty index is linked to cause-specific deaths, particularly deaths from cancer, in CKD population in the United States (US). Therefore, we conducted a cohort study to assess the relationship of the frailty index with deaths from all causes or specific causes in US adults with CKD. The present study aimed to determine the prognostic value of frailty index in CKD. We anticipate that the insights obtained from this study will be valuable for CKD treatment.

## Methods

### Study design

We used information from the "National Health and Nutrition Examination Survey" (NHANES) (1999–2018). With assistance from the National Center for Health Statistics (NCHS), the US conducted a national survey known as the NHANES. The present study constitutes a secondary analysis of data from the NHANES. All original data collection protocols were approved by the Research Ethics Review Board of the NCHS. Written informed consent was obtained from every participant at the time of the survey. Consequently, no additional ethical approval was required for the present secondary analysis, and the study fully complies with the ethical principles of the Declaration of Helsinki. The results of the study were reported according to the STrengthening the Reporting of OBservational studies in Epidemiology (STROBE) guidelines.

### Study population

The participant inclusion criteria were as follows: (1) CKD, marked by <60 mL/min/1.73 m² "estimated glomerular filtration rate" (eGFR), a urinary albumin: creatinine ratio (ACR) ≥ 30 mg/g, or both conditions [14–16] and (2) age ≥ 20 years. The exclusion criteria were as follows: (1) follow-up loss, (2) no frailty score information, (3) incomplete covariate statistics, and (4) gestation. In total, we screened 55,081 people aged at least 20 years from the NHANES database between 1999 and 2018. Of these, 9509 individuals had CKD. After excluding participants according to the exclusion criteria, we finally examined the data of 3262 participants who fulfilled the selection criteria (Fig 1).

### Frailty index

The frailty index was derived following Searle et al.'s methodology, which incorporates deficiencies across various body systems. We used a model consisting of 53 deficiencies spanning seven distinct domains, with severity ratings assigned

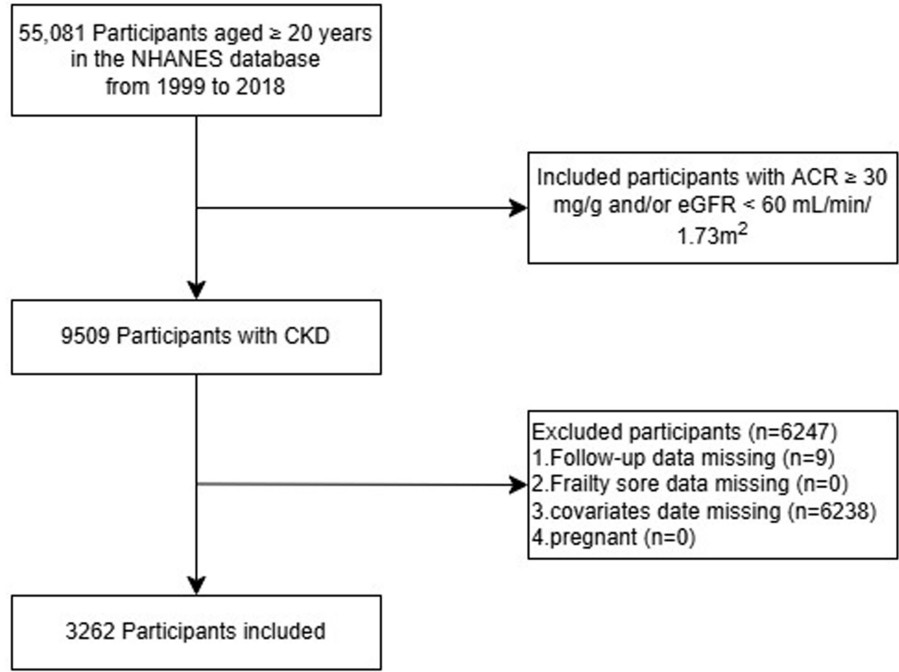

**Fig 1. Study flow chart.**

on a scale ranging from 0 to 1. The seven domains were as follows: (1) cognition (presence of issues related to consciousness and memory); (2) dependence (20 questions addressing challenges encountered in performing ADL; (3) depression (the Patient Health Questionnaire-9's seven questions); (4) comorbidities (13 questions pertaining to self-reported ailments, including thyroid disorders, arthritis, chronic bronchitis, heart failure, various malignancies, angina, coronary heart disease, myocardial infarction, hypertension, cerebrovascular accidents, diabetes mellitus, urinary incontinence, and kidney failure); (5) using hospital care and admission to healthcare facilities (five questions regarding the overall condition, a comparison of current health status to that from 1 year ago, occurrence of hospital admission throughout the night in the last year, care service numbers used over the last year, and total count of prescribed medicines); (6) physical capability and anthropometric measurements (body mass index [BMI] had one item); and (7) laboratory measurements (six items related to glycated hemoglobin [HbA1c] level, hemoglobin concentration, red blood cells, lymphocyte proportion, red cell distribution, and percentage of segmented neutrophils). The ratio of gathered deficits to total potential deficits enabled to calculate the frailty index [12,17,18]. We used this index to assign patients to tertiles 1 (0.010–0.128), 2 (0.128–0.210), and 3 (0.210–0.665).

## Mortality data

The NHANES database is linked with the Centers for Disease Control and Prevention's "national mortality index." Mortality data through December 31, 2018, were included. We considered any cause of death as all-cause mortality. The tenth revision of the International Classification of Diseases (ICD-10) was used to categorize specific death causes. We evaluated deaths from all causes alongside certain causes, such as cancer (ICD-10: C00-C97) and cardiovascular diseases (ICD-10: I00-I09, I11, I13, I20-I51). We calculated the survival time using NHANES information as our starting point.

## Covariates

Information regarding BMI, age, race, sex, marital status, educational level, poverty-to-income ratio (PIR), smoking habits, alcohol drinking, duration of overall physical activity, hypertension, and diabetes mellitus was collected from participant interviews conducted at their households. Details regarding C-reactive protein (CRP) level, urinary albumin: creatinine ratio (ACR), and the eGFR were obtained from laboratory tests. The specific definitions for these items used in the present study are detailed herein. Age was considered a continuous factor, reflecting the participant's age. Sex was documented as indicated. Race or ethnicity covered non-Hispanic Black, non-Hispanic White, other Hispanic, Mexican–American, and other groups. Marital standing was classified as unmarried, married, cohabiting, and other (which included individuals who were divorced, widowed, or separated). BMI was a continuous factor, derived from the weight and height measurements of the participants. The PIR was segmented into ranges of 1–1.3, 1.31–3.50, and >3.50. Educational attainment was classified as high school or corresponding, less than high school, and beyond high school. Smoking habit covered never-smoker (defined as having smoked <100 cigarettes ever), former smoker (≥100 cigarettes but not smoking at present), and current smoker (≥100 cigarettes and currently smoking, either rarely or every day) [19]. Alcohol drinking was categorized as never-drinker (<12 drinks ever), former drinker (minimum 12 drinks per year but not in the last year), and present drinkers [20]. We calculated the total time spent on physical activity (a continuous variable) as the weekly hours dedicated to cycling, walking, job-related activities, and free-time pursuits [21]. We defined hypertension as an average diastolic blood pressure ≥ 90 mmHg, systolic blood pressure ≥ 140 mmHg, self-admitted diagnosis, or use of blood pressure medicines. Clinicians diagnosed diabetes based on test results for a 2-h or random oral glucose tolerance test (≥11.1 mmol/L), HbA1c level ≥ 6.5% or higher fasting blood glucose of ≥7.0 mmol/L, or under insulin or diabetes treatment. Hyperlipidemia was diagnosed when a person met minimum one of these conditions: (1) taking lipid-decreasing medicine; (2) triglyceride levels ≥150 mg/dL; or (3) high cholesterol level [low-density lipoprotein ≥ 130 mg/dL, total cholesterol ≥ 200 mg/dL, or high-density lipoprotein < 40 mg/dL].

## Statistical analysis

Based on data distribution normality, number (percentage) was used to express categorical factors, and median (inter-quartile range [IQR]) or mean ± standard deviation was used to represent continuous factors. Frailty was expressed by both tertile stratification and as a natural log-transformed continuous metric. We conducted the χ² test (categorical factors), Kruskal–Wallis test (factors with non-normal distribution), and one-way analysis of variance (factors with normal distribution) to compare groups. The link between death risk from specific and all causes and frailty was determined by multivariable Cox proportional hazards regression and Kaplan–Meier estimates. Four hierarchical models were used for this test. We modified Model 1 to account for biological sex, race/ethnicity, and age (which was continuously modeled). Model 2 accounted for BMI (continuous), marital status, educational attainment, PIR category, total physical activity duration (continuous), smoking history, and alcohol drinking. Model 3 was adjusted for the status of hyperlipidemia, diabetes, and hypertension (categorical). For refining model 4, we finally included CRP, ACR, and eGFR as variables.

Restricted cubic spline was utilized to examine the dose–response relationship of the frailty index with death from all and specific causes. The analyses were stratified according to various factors, including sex (male and female), ethnicity (Caucasian or non-Caucasian), age (<65 years and ≥65 years), BMI (<30 and ≥30 kg/m²), diabetes, hypertension, and eGFR. The significance of the interactions was determined using the interaction term's P-value involving all stratified variables together with frailty scores. To validate the robustness of our results, a series of sensitivity analyses were conducted. Initially, we excluded individuals who had cardiovascular diseases. Next, individuals with a history of cancer were excluded. Finally, multiple imputation was used to address any missing data. These strategies improved the strength and dependability of our primary discoveries by re-evaluating all models that linked the frailty index to death risk.

All statistical analyses were conducted using Free Statistics software (version 2.0) and R program (version 4.2.2) (https://www.r-project.org/). Differences with a two-sided P-value of <0.05 were considered significant.

## Results

### Participant features

Table 1 presents the demographic and baseline traits of the study population, which comprised 3262 individuals diagnosed with CKD. Participants in the higher frailty index group (tertile 3) were more frequently older and had a higher BMI, higher CRP level, higher ACR, and a higher prevalence of hypertension and diabetes mellitus, together with a reduced total duration of physical activity and eGFR (all P < 0.001).

### Follow-up results

Over the median follow-up period of 9.4 years, 1102 deaths from all causes were recorded, representing 33.78% of the studied population. This included 402 cardiovascular-related deaths (12.32%) and 196 cancer-related deaths (6.00%). The observed rates of all-cause mortality across frailty index tertiles 1, 2, and 3 were 8.5%, 11.1%, and 14.2%, respectively. Furthermore, across these tertiles, the respective rates of cardiovascular mortality were 2.8%, 4.2%, and 5.4%, while those of cancer-related deaths were 1.7%, 2.1%, and 2.2%.

### Survival analysis

Fig 2 shows the Kaplan–Meier survival curves. Frailty scores were negatively linked to survival likelihood, covering death from all causes (Fig 2A), CVD mortality (Fig 2B), and cancer mortality (Fig 2C). Notably, the highest frailty index tertile (tertile 3) consistently exhibited the lowest survival probability, with statistically significant differences compared to the other frailty index tertiles (death from all causes and CVD: P < 0.0001; cancer mortality: P = 0.0026).

In the multivariable Cox regression analysis (Table 2), compared to tertile 1, the adjusted hazard ratio (HR) for death from all reasons in tertile 2 was 1.25 (1.07–1.47), while in tertile 3, the adjusted HR was 1.81 (1.54–2.13) (P$_{trend}$ < 0.001).

**Table 1. Demographic and baseline traits grouped by frailty scores.**

| Variables | Total | Frailty score | | | P value |
|---|---|---|---|---|---|
| | | Tertile 1 (0.010–0.128) | Tertile 2 (0.128–0.210) | Tertile 3 (0.210–0.665) | |
| Number of participants | 3262 | 1087 | 1087 | 1088 | |
| Age (years), mean±SD | 61.9±17.1 | 57.6±19.2 | 62.8±16.4 | 65.3±14.3 | <0.001 |
| Sex, n (%) | | | | | 0.481 |
| Male | 1715 (52.6) | 582 (53.5) | 577 (53.1) | 556 (51.1) | |
| Female | 1547 (47.4) | 505 (46.5) | 510 (46.9) | 532 (48.9) | |
| Race, n (%) | | | | | <0.001 |
| Non-Hispanic White | 1775 (54.4) | 630 (58) | 580 (53.4) | 565 (51.9) | |
| Non-Hispanic Black | 600 (18.4) | 147 (13.5) | 226 (20.8) | 227 (20.9) | |
| Mexican–American | 495 (15.2) | 182 (16.7) | 148 (13.6) | 165 (15.2) | |
| Other Hispanic | 187 (5.7) | 60 (5.5) | 64 (5.9) | 63 (5.8) | |
| Other race (including multi-racial) | 205 (6.3) | 68 (6.3) | 69 (6.3) | 68 (6.2) | |
| Marital status, n (%) | | | | | <0.001 |
| Married/Living with partner | 1894 (58.1) | 667 (61.4) | 647 (59.5) | 580 (53.3) | |
| Never married/Other | 1368 (41.9) | 420 (38.6) | 440 (40.5) | 508 (46.7) | |
| BMI (kg/m$^2$), mean±SD | 29.6±6.8 | 28.1±6.1 | 29.5±6.6 | 31.0±7.3 | <0.001 |
| PIR group, n (%) | | | | | <0.001 |
| ≤1.30 | 949 (29.1) | 279 (25.7) | 275 (25.3) | 395 (36.3) | |
| 1.31–3.50 | 1385 (42.5) | 445 (40.9) | 470 (43.2) | 470 (43.2) | |
| >3.50 | 928 (28.4) | 363 (33.4) | 342 (31.5) | 223 (20.5) | |
| Education level, n (%) | | | | | <0.001 |
| Less than high school | 917 (28.1) | 259 (23.8) | 299 (27.5) | 359 (33.0) | |
| High school or equivalent | 831 (25.5) | 283 (26) | 271 (24.9) | 277 (25.5) | |
| Above high school | 1514 (46.4) | 545 (50.1) | 517 (47.6) | 452 (41.5) | |
| Smoking status, n (%) | | | | | <0.001 |
| Never | 1571 (48.2) | 589 (54.2) | 526 (48.4) | 456 (41.9) | |
| Former | 1127 (34.5) | 312 (28.7) | 391 (36) | 424 (39) | |
| Current | 564 (17.3) | 186 (17.1) | 170 (15.6) | 208 (19.1) | |
| Physical activity duration, median (IQR), min/week | 225.0(79.1, 660.0) | 240.0(90.0, 720.0) | 236.2(90.0, 690.0) | 204.8(63.0, 600.0)) | 0.004 |
| Alcohol intake, n (%) | | | | | <0.001 |
| Never | 462 (14.2) | 155 (14.3) | 149 (13.7) | 158 (14.5) | |
| Former | 876 (26.9) | 213 (19.6) | 277 (25.5) | 386 (35.5) | |
| Current | 1924 (59.0) | 719 (66.1) | 661 (60.8) | 544 (50) | |
| Hypertension, n (%) | | | | | <0.001 |
| No | 1061 (32.5) | 551 (50.7) | 330 (30.4) | 180 (16.5) | |
| Yes | 2201 (67.5) | 536 (49.3) | 757 (69.6) | 908 (83.5) | |
| Diabetes, n (%) | | | | | <0.001 |
| No | 2155 (66.1) | 926 (85.2) | 710 (65.3) | 519 (47.7) | |
| Yes | 1107 (33.9) | 161 (14.8) | 377 (34.7) | 569 (52.3) | |
| Hyperlipidemia, n (%) | | | | | <0.001 |
| No | 596 (18.3) | 258 (23.7) | 209 (19.2) | 129 (11.9) | |
| Yes | 2666 (81.7) | 829 (76.3) | 878 (80.8) | 959 (88.1) | |
| CRP (mg/L), median (IQR) | 2.5(1.1,5.7) | 1.9(0.9,4.2) | 2.5(1.1,5.5) | 3.2(1.4,7.4) | <0.001 |

*(Continued)*

**Table 1.** (Continued)

| Variables | Total | Frailty score | | | P value |
|---|---|---|---|---|---|
| | | Tertile 1 (0.010–0.128) | Tertile 2 (0.128–0.210) | Tertile 3 (0.210–0.665) | |
| ACR (mg/g), median (IQR) | 43.0(13.6,97.4) | 40.0(14.8,76.9) | 43.3(13.5,96.9) | 48.2(13.1,126.8) | <0.001 |
| eGFR (mL/min/1.73 m²), mean ± SD | 73.5 ± 28.0 | 80.3 ± 27.3 | 73.8 ± 27.8 | 66.3 ± 27.1 | <0.001 |

IQR, interquartile range; BMI, body mass index; SD, standard deviation; PIR, poverty-to-income ratio; CRP, C-reactive protein; ACR, urinary albumin: Creatinine ratio; eGFR, estimated glomerular filtration rate.

The adjusted HR values for cardiovascular mortality in tertiles 2 and 3 were 1.44 (1.09–1.89) and 2.00 (1.52–2.64) ($P_{trend}$<0.001), respectively. The adjusted HR for cancer mortality in tertile 1 vs. tertile 2 was 1.14 (0.79–1.64), and the adjusted HR in tertile 3 (vs. tertile 1) was 1.59 (1.09–2.33) (P=0.017). Furthermore, for every increase in log-transformed frailty index by one unit, we found a notable upsurge in mortality. Specifically, death from all causes, cardiovascular illnesses, and cancer increased by 85% (P<0.001), 112% (P<0.001), and 72% (P<0.001), respectively.

### Restricted cubic spline analysis and stratified analysis

Major dose–response relationships were observed between the frailty index and the risk of deaths from all causes, cancer, and cardiovascular illnesses (Fig 3). An increased risk for all three forms of death was linked to a greater frailty index. Additionally, the stratified analysis confirmed a positive correlation between the frailty index and death risk across several subgroups, including sex, hypertension, age, BMI, race, diabetes, and eGFR (Fig 4).

### Sensitivity analysis

Table 3 shows the results of sensitivity analysis to evaluate the reliability of the main analysis conducted using multivariable Cox regression statistics (Model 4). Following the exclusion of individuals with a history of cardiovascular illness, the adjusted HR values for tertiles 2 and 3 were 1.21 (1.00–1.48) and 1.69 (1.38–2.06), respectively, compared to tertile 1 ($P_{trend}$<0.001). An increase in the log-transformed frailty index by one unit suggested a 69% increase in death from all causes (P<0.001). Additionally, following the exclusion of individuals with a history of cancer, the adjusted HR values for tertiles 2 and 3 were 1.50 (1.24–1.80) and 2.01 (1.66–2.43), respectively, compared to tertile 1 ($P_{trend}$<0.001). Additionally, an increase of 1 unit in the log-transformed frailty index was linked to a 91% increase in death risk from all causes (P<0.001). In the analysis of missing data through multiple imputation, the adjusted HR (95% CI) values for tertiles 2 and 3 were 1.38 (1.26–1.51) and 2.47 (2.25–2.71), respectively, compared to tertile 1 ($P_{trend}$<0.001). Furthermore, an increase of 1 unit in the log-transformed frailty index corresponded to a substantial 119% increase in death from all causes (P<0.001).

### Discussion

The present study revealed that, in CKD patients from the US, the frailty index was significantly linked to death from all causes and specific causes. In particular, a greater frailty index was closely associated with an increased risk of death from all causes, heart disease, and cancer. In CKD treatment, our results highlight the prognostic value of frailty tests, supporting its incorporation into treatment planning.

Our findings regarding the link between death and frailty are consistent with those of previous studies. A prospective, observational study of frailty, quality of life, and dialysis in older people with advanced CKD [22] showed that for every 0.1 baseline frailty index increase, the death risk increased by 59%; frailty significantly increased over a 4-year period and was linked to higher death rates. As reported by Kennard et al. [23], frailty prevalence was closely associated with survival

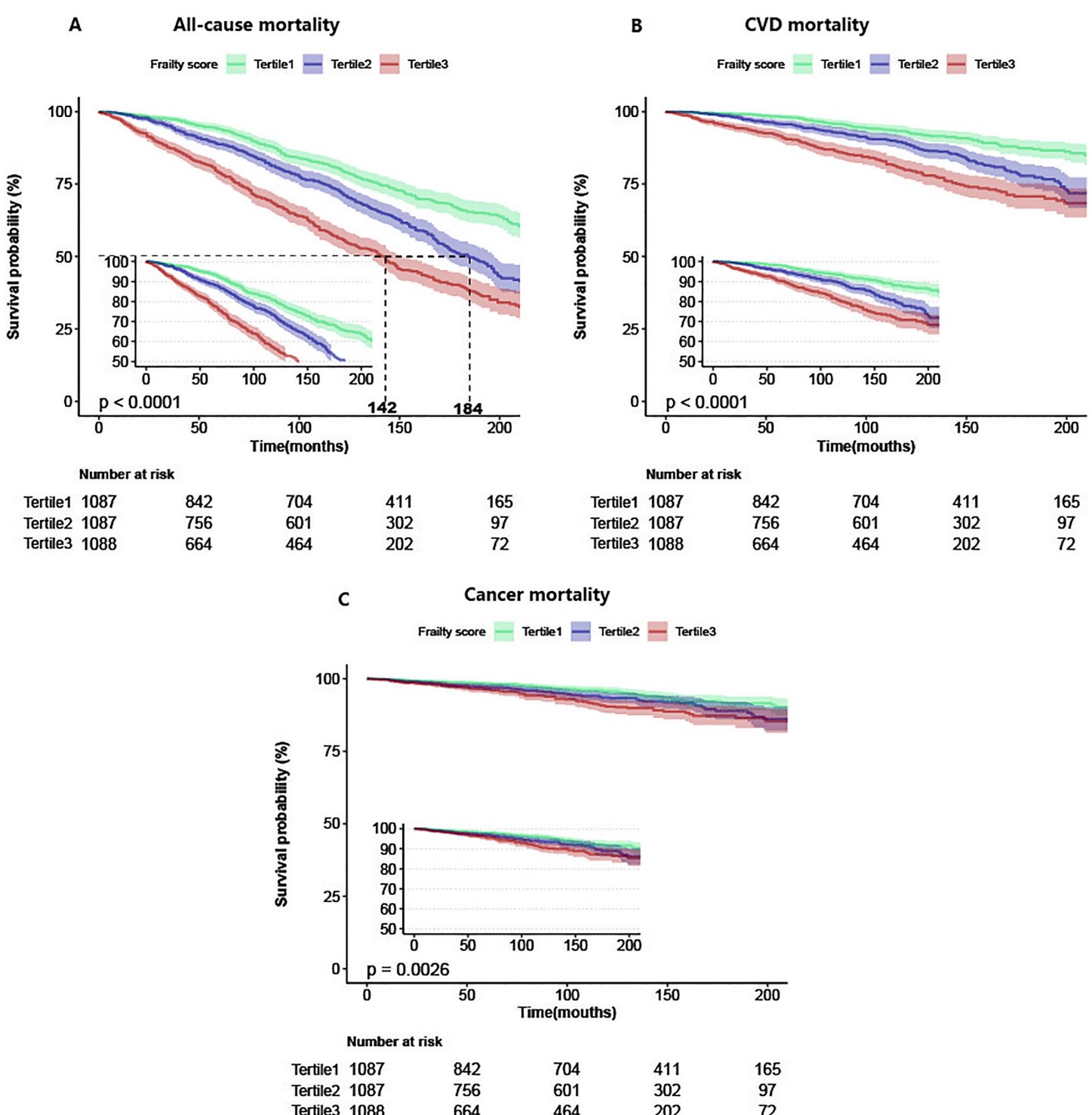

**Fig 2. Kaplan–Meier survival curves depicting death from all causes (A), cardiovascular illnesses (B), and cancer (C) stratified by tertiles of frailty scores.**

**Table 2. Multivariate analysis of the link between the frailty score and mortality.**

| Characteristics | Frailty score | | | P_trend | Per unit increase in the frailty score (log-transformed) |
|---|---|---|---|---|---|
| | Tertile 1 (0.010–0.128) | Tertile 2 (0. 128–0.210) | Tertile 3 (0.210–0.665) | | |
| All-cause mortality | | | | | |
| Model 1 | Reference | 1.13 (1.11–1.52) | 2.14 (1.84–2.49) | <0.001 | 2.08 (1.84–2.35) |
| Model 2 | Reference | 1.36 (1.16–1.59) | 2.11 (1.81–2.46) | <0.001 | 2.05 (1.81–2.32) |
| Model 3 | Reference | 1.26 (1.08–1.48) | 1.87 (1.60–2.20) | <0.001 | 1.89 (1.65–2.15) |
| Model 4 | Reference | 1.25 (1.07–1.47) | 1.81 (1.54–2.13) | <0.001 | 1.85 (1.62–2.12) |
| CVD mortality | | | | | |
| Model 1 | Reference | 1.49 (1.14–1.94) | 2.5 (1.93–3.24) | <0.001 | 2.50 (2.02–3.09) |
| Model 2 | Reference | 1.57 (1.20–2.05) | 2.36 (1.82–3.07) | <0.001 | 2.36 (1.91–2.93) |
| Model 3 | Reference | 1.45 (1.10–1.90) | 2.06 (1.57–2.71) | <0.001 | 2.15 (1.72–2.69) |
| Model 4 | Reference | 1.44 (1.09–1.89) | 2.00 (1.52–2.64) | <0.001 | 2.12 (1.68–2.66) |
| Cancer mortality | | | | | |
| Model 1 | Reference | 1.13 (0.79–1.62) | 1.60 (1.12–2.27) | 0.009 | 1.68 (1.26–2.23) |
| Model 2 | Reference | 1.14 (0.79–1.63) | 1.59 (1.10–2.28) | 0.012 | 1.68 (1.25–2.25) |
| Model 3 | Reference | 1.14 (0.79–1.64) | 1.62 (1.11–2.36) | 0.012 | 1.74 (1.28–2.37) |
| Model 4 | Reference | 1.14 (0.79–1.64) | 1.59 (1.09–2.33) | 0.017 | 1.72 (1.26–2.35) |

Model 1: adjusted for age, race, and sex.

Model 2: Model 1 plus adjustment for marital status, PIR category, educational level, smoking habit, duration of physical activity, BMI, and alcohol drinking.

Model 3: Model 2 plus adjustment for diabetes, hypertension, and hyperlipidemia.

Model 4: Model 3 plus adjustment for CRP, ACR, and eGFR.

PIR, poverty-to-income ratio; BMI, body mass index; CRP, C-reactive protein; ACR, urinary albumin: creatinine ratio; eGFR, estimated glomerular filtration rate.

outcomes. In their study of 256 participants, 36.3% and 46.5% of participants exhibited frailty and pre-frailty, respectively. The predictive value of frailty status for death risk (HR 2.83, 95% CI 1.44–5.56) was superior to that of age, comorbidities, and laboratory indicators. Previous meta-analyses and systematic reviews also support the link of frailty with death outcomes in people with CKD [24–27].

Our findings suggest that frailty in people with CKD considerably influences not only general mortality but also mortality associated with cardiovascular diseases and cancer. This finding is consistent with the results of Lohman et al. [28], who showed that the all-cause mortality risk of older individuals with frailty was significantly higher than that of asymptomatic individuals (adjusted HR 2.75, 95% CI 2.14–3.53). Frailty was associated with an increased risk of mortality from heart disease, cancer, respiratory diseases, and dementia by 2.96-fold (95% CI 2.17–4.03), 2.82-fold (95% CI 2.02–3.94), 3.48-fold (95% CI 2.17–5.59), and 2.87-fold (95% CI 1.47–5.59), respectively. According to our subgroup analysis, the frailty index was linked to death risk in the CKD population in specific age groups, sex, BMI categories, races, renal function status, and diabetes and hypertension status. Individuals under 65 years of age showed a stronger link between death and frailty. This finding supports the results of a Swedish cohort research [29], which showed that the frailty index has a higher predictive power in younger people and is a useful indicator of death. Wilkinson et al. [10] also demonstrated that a higher death risk among people with CKD is linked to frailty, even after adjusting for other covariates. Their findings showed that death rates in patients with CKD increased by 60% to 116% in the presence of moderate-to-severe frailty. Our sensitivity analysis revealed that the link of the frailty index with mortality was still statistically significant after excluding participants with documented CVD or cancer history. Collectively, these findings imply that frailty is a robust and accurate indicator of death risk in the CKD population.

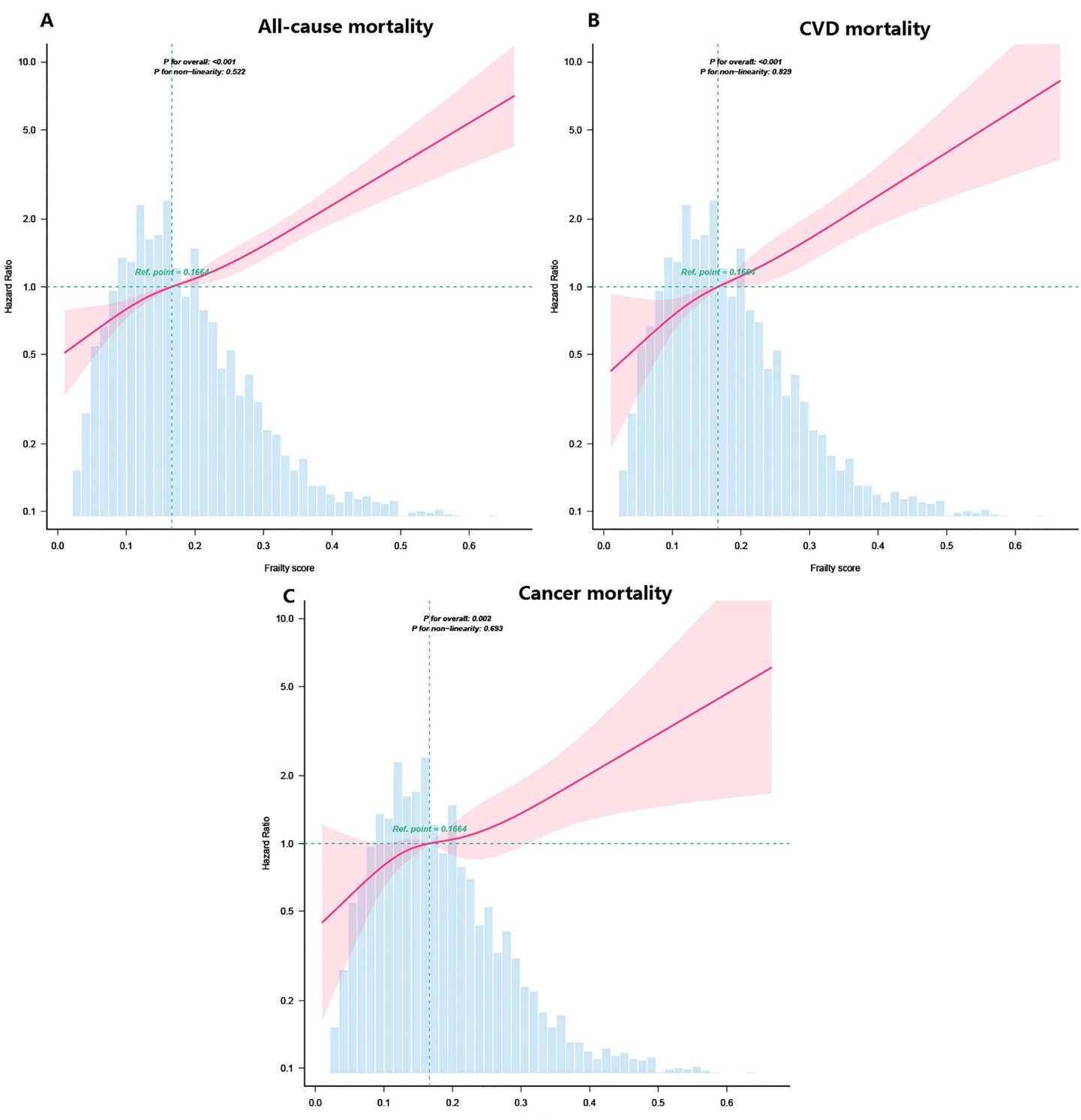

**Fig 3. Link of frailty scores with death from all causes (A), CVD (B), and cancer (C) in the dose–response analysis.** The restricted cubic spline analysis has been modified for race, sex, age, BMI, marital status, PIR, educational attainment, smoking habits, duration of physical activity, and alcohol consumption, along with hypertension, diabetes, hyperlipidemia, CRP, ACR, and eGFR. CKD, chronic kidney disease; BMI, body mass index; PIR, poverty-to-income ratio; CRP, C-reactive protein; ACR, urinary albumin: creatinine ratio; eGFR, estimated glomerular filtration rate.

| Subgroup | Total | Event (%) | HR (95%CI) | All-cause mortality | P for interaction |
|---|---|---|---|---|---|
| **Overall** | | | | | |
| Crude | 3262 | 1102(33.8) | 2.27(2.02~2.54) | | |
| Adjusted | 3262 | 1102(33.8) | 1.85(1.62~2.12) | | |
| **Age** | | | | | **0.001** |
| < 65 years | 1550 | 214 (13.8) | 2.14 (1.57~2.92) | | |
| ≥ 65 years | 1712 | 888 (51.9) | 1.73 (1.49~2.01) | | |
| **Sex** | | | | | **0.584** |
| Male | 1715 | 679 (39.6) | 1.78 (1.49~2.13) | | |
| Female | 1547 | 423 (27.3) | 1.93 (1.57~2.38) | | |
| **Race** | | | | | **0.793** |
| White | 1775 | 768 (43.3) | 1.84 (1.56~2.16) | | |
| Othes | 1487 | 334 (22.5) | 1.87 (1.47~2.38) | | |
| **BMI** | | | | | **0.307** |
| < 30kg/m² | 1934 | 757 (39.1) | 1.83 (1.56~2.14) | | |
| ≥ 30kg/m² | 1328 | 345 (26) | 2 (1.53~2.6) | | |
| **Hypertension** | | | | | **0.806** |
| No | 1061 | 224 (21.1) | 1.71 (1.29~2.28) | | |
| Yes | 2201 | 878 (39.9) | 1.91 (1.63~2.23) | | |
| **Diabetes** | | | | | **0.253** |
| No | 2155 | 703 (32.6) | 1.74 (1.48~2.03) | | |
| Yes | 1107 | 399 (36) | 2.17 (1.67~2.82) | | |
| **eGFR (mL/min/1.73m²)** | | | | | **0.968** |
| ≥ 90 | 1015 | 132 (13) | 1.68 (1.15~2.44) | | |
| [60,90) | 764 | 280 (36.6) | 1.72 (1.32~2.24) | | |
| [30,60) | 1383 | 627 (45.3) | 1.83 (1.53~2.19) | | |
| < 30 | 100 | 63 (63) | 3.38 (1.32~8.65) | | |

Effect (95%CI): 1.0  2.0  4.0  8.0

**Fig 4. Subgroup analyses results show the link of the frailty index with death from all causes.** In the Cox proportional hazards regression analysis, HR and 95% CI values were calculated. The model was adjusted for age, PIR, marital status, sex, BMI, race, smoking habits, educational level, alcohol drinking, time spent in physical activity, CRP, ACR, eGFR, and comorbidities (including hyperlipidemia, diabetes, and hypertension). CI, confidence interval; BMI, body mass index; HR, hazard ratio; CKD, chronic kidney disease; PIR, poverty-to-income ratio; CRP, C-reactive protein; ACR, urinary albumin: creatinine ratio; eGFR, estimated glomerular filtration rate.

**Table 3. Results of the sensitivity analysis.**

| Characteristics | Frailty score | | | P$_{trend}$ | Per-unit increase in frailty scores (log-transformed) |
|---|---|---|---|---|---|
| | Tertile 1 | Tertile 2 | Tertile 3 | | |
| Excluding participants with a history of CVD | | | | | |
| Model 1 | Reference | 1.23 (1.02–1.50) | 1.92 (1.60–2.32) | <0.001 | 1.89 (1.62–2.21) |
| Model 2 | Reference | 1.29 (1.06–1.57) | 1.90 (1.57–2.29) | <0.001 | 1.89 (1.62–2.21) |
| Model 3 | Reference | 1.22 (1.00–1.48) | 1.71 (1.40–2.08) | <0.001 | 1.76 (1.49–2.07) |
| Model 4 | Reference | 1.21 (1.00–1.48) | 1.69 (1.38–2.06) | <0.001 | 1.69 (1.38–2.06) |
| Excluding participants with a history of cancer | | | | | |
| Model 1 | Reference | 1.50(1.25–1.79) | 2.34 (1.96–2.78) | <0.001 | 2.09 (1.83–2.39) |
| Model 2 | Reference | 1.60 (1.33–1.92) | 2.32 (1.94–2.77) | <0.001 | 2.08 (1.81–2.38) |
| Model 3 | Reference | 1.50 (1.25–1.81) | 2.07 (1.71–2.50) | <0.001 | 1.94 (1.67–2.24) |
| Model 4 | Reference | 1.50 (1.24–1.80) | 2.01 (1.66–2.43) | <0.001 | 1.91 (1.64–2.22) |
| Multiple imputation of missing data | | | | | |
| Model 1 | Reference | 1.45 (1.33–1.59) | 2.87 (2.64–3.12) | <0.001 | 2.42 (2.27–2.58) |
| Model 2 | Reference | 1.44 (1.32–1.58) | 2.74 (2.51–3.00) | <0.001 | 2.35 (2.20–2.52) |
| Model 3 | Reference | 1.40 (1.28–1.54) | 2.61 (2.38–2.86) | <0.001 | 2.29 (2.14–2.46) |
| Model 4 | Reference | 1.38 (1.26–1.51) | 2.47 (2.25–2.71) | <0.001 | 2.19 (2.04–2.35) |

Model 1: adjusted for age, race, and sex.

Model 2: Model 1 plus adjustment for marital status, PIR category, BMI, smoking habit, educational level, time spent in physical activity, and alcohol drinking.

Model 3: Model 2 plus adjustment for diabetes, hypertension, and hyperlipidemia.

Model 4: Model 3 plus adjustment for CRP, ACR, and eGFR.

PIR, poverty-to-income ratio; BMI, body mass index; CRP, C-reactive protein; ACR, urinary albumin: creatinine ratio; eGFR, estimated glomerular filtration rate.

A key discovery in the present study is the dose–response association between the frailty index and the likelihood of death from all causes, cancer, and heart diseases in individuals with CKD. The highest tertile had a 181% greater risk of death from all causes, a 159% greater risk of death from cancer, and a 200% greater risk of death from heart disease when compared with the lowest frailty index tertile. Our results align with those of Lohman et al., who reported the link of frailty with more deaths in older adults [28].

We observed that the CKD population with a high frailty index generally comprised individuals who consumed alcohol, had a high BMI, and had a shorter duration of physical activity. A substantial body of research has established that various elements, including dietary intake, lifestyle behaviors, comorbidities, and psychological influence, play significant roles in the onset and advancement of frailty [30–32]. These varying features indicate that frailty is a revocable, multifaceted condition. Consequently, it is imperative to implement comprehensive preventive or interventional strategies to address the issue of frailty in patients with CKD. Although existing research indicates that multidimensional interventions, such as nutritional support, physical therapy, and psychological interventions, significantly improve the frailty status of patients with CKD [33], further exploration is required to determine the optimal approach in clinical practice based on the perspectives and findings of different studies.

The vulnerable condition of patients with CKD may elevate the mortality risk through multiple biological pathways. Both CKD and frailty are linked to persistent inflammatory responses, which can increase the likelihood of developing cardio-vascular conditions and cancer [34]. Moreover, individuals with CKD frequently undergo lifestyle modifications, including the adoption of poor dietary habits and reduced physical activity. These changes can further intensify frailty and elevate the mortality risk [35,36]. Frailty also has the potential to impact cognitive ability and the capacity to perform ADL, which can render individuals more vulnerable to unintentional injuries and infections, consequently increasing the likelihood of

mortality [37]. Finally, the tolerance of frail patients with malignant tumors to chemotherapy is significantly limited, and they experience a higher risk of treatment-related toxicity, leading to a substantially increased mortality rate [38–40].

## Study strengths

This study has multiple strengths that are worthy of mention. First, we performed a comprehensive, large-scale study of the link of frailty index with death risk in US adults with CKD. Different age groups were analyzed, which can improve our understanding of the frailty condition in the CKD population with diverse ages and how it affects all-cause death and cause-specific mortality risk, providing more thorough insights for clinical application. Second, the analysis was based on data from a nationally representative cohort derived from the NHANES, which improved the external validity and generalizability of our results. Finally, we implemented robust statistical methodologies, including sensitivity analyses and multivariable adjustments, to ascertain the strength of our conclusions.

## Study limitations

When interpreting our results, it is important to acknowledge the limitations of our study. First, the external validity of our results may have been impacted by selection bias due to our reliance on retrospective data. Second, we calculated the frailty index using self-admitted information from all patients, which could have potentially resulted in self-reporting bias. In addition, although we utilized an optimal strategy to adjust for likely confounding aspects, it is possible that unassessed or unknown confounding variables remained, such as CKD treatment variable, which were not adjusted. Finally, the observational research design prevented us from determining causative associations.

In light of the limitations of this study, it will be essential for future studies to include unassessed variables such as treatment variables to provide a more comprehensive understanding of frailty status in the CKD population.

## Conclusion

In conclusion, this study revealed a significant link between the frailty index and death from all causes and specific causes in US adults with CKD, underscoring the potential utility of this index as a prognostic tool. To enhance the quality of life and prognosis for the CKD population, clinical practice should integrate frailty management with patient care.

## Supporting information

**S1 Data. Data.**
(CSV)

## Author contributions

**Conceptualization:** Xiaoyan Lu.

**Data curation:** Mengmei Xiong, Xiaoyan Lu.

**Formal analysis:** Mengmei Xiong, Xiaoyan Lu.

**Investigation:** Mengmei Xiong.

**Methodology:** Xiaoyan Lu.

**Supervision:** Mengmei Xiong.

**Validation:** Mengmei Xiong, Xiaoyan Lu.

**Visualization:** Mengmei Xiong.

**Writing – original draft:** Xiaoyan Lu.

**Writing – review & editing:** Mengmei Xiong.

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
