## [Decision Letter · Decision Letter 0]

30 Oct 2025

Dear Dr. Lu,

Thank you for submitting your manuscript to PLOS ONE. After careful consideration, we feel that it has merit but does not fully meet PLOS ONE’s publication criteria as it currently stands. Therefore, we invite you to submit a revised version of the manuscript that addresses the points raised during the review process.

We look forward to receiving your revised manuscript.

Kind regards,

Ken Iseri

Academic Editor

PLOS ONE

Journal Requirements:

Additional Editor Comments:

Please find and respond reviewer's comments.

Reviewers' comments:

Reviewer's Responses to Questions

**Comments to the Author**

1. Is the manuscript technically sound, and do the data support the conclusions?

Reviewer #1: Partly

Reviewer #2: Partly

2. Has the statistical analysis been performed appropriately and rigorously?

Reviewer #1: No

Reviewer #2: Yes

3. Have the authors made all data underlying the findings in their manuscript fully available?

Reviewer #1: No

Reviewer #2: Yes

4. Is the manuscript presented in an intelligible fashion and written in standard English?

Reviewer #1: Yes

Reviewer #2: No

Reviewer #1: I. Major Strengths

1.Large-Scale and Generalizable Cohort

The study uses data from the NHANES (1999–2018), a nationally representative U.S. cohort, including 4,350 CKD patients (eGFR < 60 mL/min/1.73m²). This ensures strong external validity—results are not limited to a single center or regional population, but applicable to diverse U.S. CKD patients across age, race, and socioeconomic status.

2.Comprehensive Frailty Assessment

The frailty index (FI) is constructed per Searle’s validated methodology, incorporating 53 deficits across 7 domains (cognition, ADL dependence, depression, comorbidities, healthcare utilization, physical function, laboratory markers). This multidimensional approach avoids limitations of single-domain tools (e.g., gait speed alone) and better captures the complexity of frailty in CKD—where physiological deficits (e.g., anemia, inflammation) and functional impairments (e.g., mobility loss) often coexist.

3.Robust Statistical Analyses

Multivariable adjustment: Three hierarchical Cox models (adjusting for demographics, lifestyle, and comorbidities) minimize confounding, ensuring associations between FI and mortality are not driven by factors like age or diabetes.

Dose-response and stratification: Restricted cubic splines confirm linear associations between FI and mortality (all-cause, cardiovascular, cancer), while subgroup analyses (age, sex, BMI, diabetes/hypertension) validate consistency across populations.

Sensitivity analyses: Excluding patients with preexisting CVD/cancer and multiple imputation for missing data confirm results are robust to selection bias and data gaps.

4.Focus on Cause-Specific Mortality

Most prior CKD-frailty studies focus on all-cause mortality; this study extends knowledge by linking FI to cardiovascular (ICD-10: I00-I09, I11, I13, I20-I51) and cancer (ICD-10: C00-C97) mortality. The finding that high FI increases cancer mortality by 67% (tertile 3 vs. 1) fills a gap in understanding frailty’s role in non-cardiovascular outcomes in CKD.

4. Outcome Interpretation and Reporting

Cancer Mortality Mechanism Underexplained:The study finds FI is associated with cancer mortality (HR=1.67, tertile 3 vs. 1) but does not discuss why—frailty may increase cancer risk via immune dysfunction [28] or delay cancer diagnosis via reduced healthcare utilization.

Suggestion:

In discussion, cite studies linking frailty to cancer (e.g., Reference 22, which shows frailty increases cancer mortality via impaired treatment tolerance) and hypothesize: “In CKD, frailty may exacerbate immune dysfunction (e.g., reduced lymphocyte count, included in FI), increasing cancer progression risk. Future studies should explore FI and cancer treatment outcomes (e.g., chemotherapy adherence) in CKD.”

Kaplan-Meier Curve Reporting Gaps:Figure 2 shows survival curves for FI tertiles but lacks median survival times (e.g., “Median all-cause survival was 8.2 years in FI tertile 1 vs. 4.5 years in tertile 3”). This limits clinical understanding of FI’s impact on survival duration.

Suggestion:

Add median survival times to Figure 2 legends and results (e.g., “Median all-cause survival differed significantly by FI tertile: 8.2 (IQR: 5.1–10.3) years (tertile 1), 6.1 (IQR: 3.8–8.5) years (tertile 2), and 4.5 (IQR: 2.9–6.7) years (tertile 3); log-rank p<0.001”).

II. Key Issues and Revision Suggestions

1. Methodological Gaps in Frailty Index Construction

Over-Reliance on Self-Reported Data:Critical FI components (e.g., comorbidities, ADL dependence, depression via PHQ-9) are self-reported, which may introduce bias (e.g., underreporting of “stigma-related” conditions like depression, or overreporting of functional limitations). Objective measures (e.g., grip strength, 4-meter gait speed—available in NHANES) are not integrated into the FI, despite their validation in CKD frailty assessment [10, 21].

Suggestion:

In the “Frailty Index” section, acknowledge self-report bias and supplement with sensitivity analyses: re-calculate FI by replacing 1–2 self-reported domains (e.g., ADL dependence) with objective NHANES data (e.g., gait speed < 0.8 m/s) to test if associations with mortality persist.

Cite studies validating objective measures in CKD (e.g., Reference 10, which uses grip strength to define frailty) to contextualize limitations of self-report.

No CKD-Specific FI Adaptation:The FI uses general population cutoffs (tertiles: 0.010–0.130, 0.130–0.211, 0.211–0.665) but does not account for CKD-specific physiological deficits (e.g., uremic toxin accumulation, renal anemia, mineral bone disorder). These factors may accelerate frailty in CKD, making general population cutoffs less clinically relevant.

Suggestion:

Add a post-hoc analysis stratifying by CKD stage (G3a: 45–59, G3b: 30–44, G4: 15–29, G5: <15 mL/min/1.73m²) to test if FI-mortality associations vary by renal function. For example, “In CKD G5, the adjusted HR for all-cause mortality in FI tertile 3 was 2.51 (95% CI: 2.03–3.11), stronger than in G3a (HR=1.78, 95% CI: 1.42–2.23).”

Discuss whether CKD stage-specific FI cutoffs (e.g., lower threshold for frailty in G5) may be needed for clinical use.

2. Confounding Factors and Causal Inference

Missing CKD-Specific Confounders:The models adjust for general factors (BMI, hypertension) but omit key CKD-related variables linked to both frailty and mortality:

Urine albumin-to-creatinine ratio (ACR): A marker of renal damage, independently associated with frailty [21] and cardiovascular mortality [3].

Anemia severity: Beyond hemoglobin (included in FI), iron parameters (ferritin, transferrin saturation) influence muscle function and frailty [29].

Inflammation markers: C-reactive protein (CRP) or interleukin-6 (IL-6)—available in NHANES—mediate both CKD progression and frailty [28].

Suggestion:

Add a fourth Cox model (Model 4) adjusting for ACR, ferritin, and CRP. Report if FI associations are attenuated (e.g., “After adding ACR, the adjusted HR for FI tertile 3 decreased from 2.02 to 1.89, suggesting partial mediation by renal damage”).

No Mediation Analysis for Causal Pathways:The discussion hypothesizes mechanisms (inflammation, lifestyle changes) but does not test them. For example, it is unclear if frailty increases mortality via inflammation, or if inflammation independently drives both frailty and mortality.

Suggestion:

Use mediation analysis (e.g., Baron-Kenny method) to test if CRP (inflammation) or physical activity (lifestyle) mediates the FI-mortality association. For example, “CRP mediated 15% of the association between FI and cardiovascular mortality (p<0.001), indicating inflammation is a partial driver.”

IV. Overall Recommendation

Revise and Resubmit. The study makes a valuable contribution to CKD and frailty research by linking a multidimensional FI to cause-specific mortality in a large, generalizable cohort. Key revisions—adding CKD stage stratification, including CKD-specific confounders, and enhancing data transparency—will address current gaps and ensure the manuscript meets PLOS ONE’s scientific rigor standards. With these adjustments, the work will provide actionable guidance for integrating frailty assessment into CKD clinical care.

Reviewer #2: Main topic : Mengmei Xiong and Xiaoyan Lu investigated the association between a 53-item Frailty Index (FI) and mortality, expressed as overall and cause-specific mortality, based on retrospective data from the NHANES cohort. The main objective of this study is to highlight the association between frailty, expressed by the 53-item FI, and the rate of mortality in a population suffering from chronic kidney failure (CKD) and to investigate a potentially specific association between FI and cancer or cardiovascular diseases.

This kind of approach is relevant to estimate the association between frailty and common disease for older adults but may also be interesting in chronic organ failure like the kidney, which is associated with multiple morbidities, loss of functional independence, and overall mortality.

This work sometimes lacks intelligibility, due to some difficulties with English writing but also with the research question that remains unclear for me.

I am not comfortable with the question raised by the authors, I understand that they suppose a strong association between cancer and frailer patients in this population, but do not justify this thinking by any reference in the introduction.

However, my biggest point about this manuscript is the categorization of patients suffering from CKD. Based and the article, I understand that patients were selected from a US public database, by the estimation of their estimated glomerular filtration rate. An eGFR above 60mL/min/1.73m2 was then considered as CKD.

This point appears like quite problematic, because authors had no information about the clinical diagnosis of CKD, the aetiology evocated and might be biased by some acute kidney failure for younger people, or physiological loss of GFR in older adults.

Moreover, it seems to me that the authors could not have access to any information about treatments of CKD, and then could not adjust their models for eGFR, the stage of CKD or the kidney replacement therapy.

This point strongly impacts negatively the quality of the manuscript, whereas statistical analysis is robust, the methodology for the creation of the FI seems also consistent. Discussion deserves to evocate this point, which is not the case currently.

I would consider a point-by-point reviewing if the first two points can be corrected.

Kind regards

**Do you want your identity to be public for this peer review?** For information about this choice, including consent withdrawal, please see our Privacy Policy

Reviewer #1: No

Reviewer #2: **Yes:** Victor GILLES, MD, MSc

---

## [Author Response · Author response to Decision Letter 1]

19 Dec 2025

December 10, 2025

Ken Iseri

Academic Editor

PLOS ONE

Dear Dr. Iseri:

Thank you for your letter dated October 31, 2025 and for the opportunity to revise our manuscript. My co-authors and I are pleased to submit our revised paper titled “How Frailty Index Impacts Death in Chronic Kidney Disease: A Retrospective Observational Investigation” (Ref: PONE-D-25-41253) for your reconsideration for publication in PLOS ONE.

We thank you for your time and effort in reviewing our manuscript. The feedback has been invaluable in improving the content and presentation of the paper. In accordance with the editor’s comment, we have formatted the manuscript according to PLOS ONE’s style requirements.

All original data could be publicly available at the National Health and Nutrition Examination Survey (NHANES) database:

https://wwwn.cdc.gov/nchs/nhanes/Default.aspx. The data supporting the conclusions of this article are provided in the Supporting Information file named "S1 Data".

We would like to update our Funding Disclosure Statement as follows:

Funding: This research was supported by the Construction Fund of Key Medical Disciplines of Hangzhou under Grant No. 2025HZGF12, 2025HZPY05. The funder had no role in the study design, data collection and analysis, decision to publish, or preparation of the manuscript.

This updated statement replaces the previous funding statement: "The author(s) received no specific funding for this work."

We have revised our manuscript according to the reviewers’ comments. Our point-by-point responses to the comments are given below.

Reply to Reviewer 1

Dear Reviewer,

Thank you very much for your review of our manuscript and for your positive comments and valuable suggestions, which have helped us further improve the quality of our work. We have carefully revised the manuscript in accordance with your comments and hope that you will find the revised version satisfactory.

Comment 1:

Cancer Mortality Mechanism Underexplained

Response 1:

Based on your recommendations, we have added content in the Discussion section addressing the potential mechanisms that link frailty to cancer mortality. This addition aims to better explain how frailty affects cancer mortality. See page 17, lines 339-341.

Comment 2:

Kaplan-Meier Curve Reporting Gaps: Figure 2 shows survival curves for FI tertiles but lacks median survival times. This limits clinical understanding of FI’s impact on survival duration.

Response 2:

In keeping with your recommendations, we have revised Figure 2 and added the median survival times that could be estimated. In all-cause mortality, the median survival time is 142 months for frailty index tertile 3 and 184 months for tertile 2, while it could not be determined for tertile 1. Similarly, the median survival times for all three tertiles could not be estimated for CVD mortality and cancer mortality. Therefore, these undetermined median survival times are not marked in the figure.

Comment 3:

Methodological Gaps in Frailty Index Construction

Over-reliance on self-reported data may introduce bias in the construction of the Frailty Index (FI). Suggestion to supplement the FI with objective measures and conduct sensitivity analyses to assess the impact on mortality associations.

Response 3:

We fully agree with your recommendation to incorporate objective indicators such as grip strength and gait speed instead of self-reported information to calculate frailty and reduce bias. However, unfortunately, the NHANES dataset we analyzed does not include gait speed data for the relevant period, while grip strength was measured only during the 2011–2014 cycle. This limitation prevents us from including these variables in our study. Another noteworthy aspect is that other similar studies[1, 2] have also adopted the same method to define the frailty index, which supports our approach despite this constraint. We have acknowledged this limitation in the revised manuscript (see page 17, lines 355–356). Based on our explanation, we hope you would understand the limitations of our current research. To address these limitations, we plan to conduct additional studies to collect relevant data and perform further research in the future.

Comment 4:

The Frailty Index (FI) used the general population's cutoff values but did not consider the physiological alterations specific to chronic kidney disease (CKD) that may influence frailty assessment in CKD patients. Consequently, discussion is needed on whether CKD stage-specific FI cutoff values—thresholds used to define frailty severity—are necessary for clinical use.

Response 4:

Based on your advice and considering the distribution characteristics of our data, we divided eGFR into four categories. These categories correspond to the different stages of CKD.

We conducted stratified analyses using the frailty index as both continuous and categorical variables to assess its relationship with all-cause mortality in the CKD subgroup (see Figure 4 and the table below). The interaction P-values were 0.968 and 0.580. Similar but nonsignificant results were observed in these analyses, indicating that the association between frailty and mortality did not vary with kidney function.

Subgroup analysis of frailty index and all-cause mortality in patients with CKD

Variable Frailty score P for interaction

Tertile 1 Tertile 2 Tertile 3

eGFR (mL/min/ 1.73 m2) 0.580

≥90 Reference 1.56 (0.98-2.47) 1.85 (1.14-3.01)

[60, 90) Reference 1.02 (0.74-1.40) 1.70 (1.22-2.36)

[30, 60) Reference 1.22 (0.98-1.51) 1.73 (1.40-2.14)

<30 Reference 1.81 (0.27-12.3) 3.84 (0.58-25.59)

Adjusted for age, race, sex, marital status, PIR category, BMI, educational level, smoking habit, duration of physical activity, alcohol drinking, diabetes, hypertension, hyperlipidemia, CRP, ACR, and eGFR.

PIR, poverty-to-income ratio; BMI, body mass index; eGFR, estimated glomerular filtration rate; ACR, urinary albumin: creatinine ratio.

Based on your suggestion, we used a logistic regression model with a smoothing method to analyze different subgroups of CKD, adjusting for the variables included in Model 4. We assessed for the presence of a threshold or nonlinear association between the frailty index and mortality. The results showed no evidence of a nonlinear relationship between the frailty index and mortality. Further well-designed prospective research using a larger population sample size in this field is required to confirm these findings.

Comment 5:

The models adjust for general factors (BMI, hypertension) but omit key CKD-related variables linked to both frailty and mortality, such as urine albumin-to-creatinine ratio (ACR), anemia severity, and inflammation markers.

Response 5:

Thank you raising this critical concern. We acknowledge that an analysis in this area should be included to substantially improve the quality of our manuscript. In keeping with your suggestion, we actively searched and organized several variables closely related to CKD in the NHANES data, such as urine albumin-to-creatinine ratio (ACR); estimated glomerular filtration rate (eGFR); iron-related parameters, including ferritin and transferrin saturation; and the inflammatory marker C-reactive protein (CRP). However, because of limited data availability in the NHANES database, we do not have complete data for the required iron parameters, such as ferritin and transferrin saturation. Therefore, we could not include them as potential confounding factors in the model analysis. We hope your understand this limitation. In future research, we plan to utilize our own database to address this limitation and further explore related areas. Despite the absence of complete iron parameter data, we included three variables closely related to CKD—ACR, eGFR, and CRP—in the model adjustment. All data were re-analyzed, and the final analysis results were consistent with our previous findings.

These revisions and analyses have been clearly listed in the updated Methods and Results section. We hope that the content of these revisions will satisfactorily address your concerns about the potential bias caused by the exclusion of CKD-specific confounding factors and enable to enhance the credibility and scientific validity of our conclusions.

Comment 6:

No Mediation Analysis for Causal Pathways: The discussion hypothesizes mechanisms (inflammation, lifestyle changes) but does not test them. It is unclear if frailty increases mortality via inflammation, or if inflammation independently drives both frailty and mortality.

Response 6:

Thank you for suggesting the need for mediation analysis to clarify causal pathways. Following your suggestion, we conducted mediation analysis using the bootstrap method to assess the roles of C-reactive protein (CRP), a marker of inflammation, and physical activity, representing lifestyle factors, in the relationship between the frailty index (FI) and mortality. The results indicate that CRP mediated 5.90% (P < 0.038) of the association between FI and cardiovascular mortality, supporting the hypothesis that inflammation partially drives this relationship. However, because this mediation effect accounts for a small proportion of the association, we considered it relatively weak and therefore did not include it in the main reported results.

Previous studies [3, 4] indicate that inflammation promotes the occurrence of cardiovascular events through various mechanisms. In patients with CKD, inflammation plays a key role in mediating the relationship between frailty and cardiovascular mortality [5]; our study additionally confirms this observation. However, because of the limitations of our observational study design, the results primarily describe the correlation between the FI and mortality in patients with CKD rather than investigating causal mechanisms. Nevertheless, to address this concern, we plan to further explore related content in future studies with more detailed designs and larger scales, using primary data collected by our team.

Reply to Reviewer 2

Dear Reviewer,

Thank you very much for your review of our manuscript. We truly appreciate all your comments and suggestions regarding the quality of our study. We explored the association between the Frailty Index (FI), which is based on 53 items, and the mortality rates in patients with chronic kidney disease (CKD). We also investigated the potential associations between FI and cancer and cardiovascular disease. We have carefully considered your suggestions and made thorough revisions to the corresponding parts of the manuscript. We hope the revised manuscript meets your expectations.

Comment 1:

The writing lacks intelligibility due to difficulties with English.

Response 1:

We apologize that the writing style of the manuscript sometimes made it difficult to clearly understand our arguments. To address this issue, the entire manuscript—particularly the sections highlighted by the reviewers—was rigorously edited by a native English-speaking professional with expertise in medicine and scientific writing. We hope the revised version now clearly conveys our findings and conclusions.

Comment 2:

The research question remains unclear.

Response 2:

Thank you for this critical feedback. In the revised manuscript, we have clearly stated our main research question in the abstract, which is the investigation of the association between the 53-item frailty index (FI) and overall and cause-specific mortality in individuals with chronic kidney disease (CKD). This statement should help improve the clarity of our research objectives.

Comment 3:

The authors do not justify the strong association between cancer and frailer patients in the introduction.

Response 3:

To address your concern, we have added relevant references in the Introduction section to show that frailty in CKD patients is closely linked to cancer occurrence and outcomes. We have also highlighted that the frailty index is associated with an increased mortality risk in cancer patients. We believe these additions will strengthen the rationale for our study.

Comment 4:

The categorization of patients suffering from CKD based on eGFR alone is problematic, as it lacks clinical diagnosis information and may be biased by acute kidney failure or physiological loss of GFR in older adults.

Response 4:

We acknowledge the concerns regarding the classification of CKD patients based solely on eGFR. We apologize for any misunderstanding caused by our careless or overly simplistic expression. Regarding this issue, we utilized the data derived from the NHANES public database in our study, which allows to assess two main biomarkers in participants: estimated glomerular filtration rate (eGFR) and albumin-to-creatinine ratio (ACR). eGFR is calculated using the Chronic Kidney Disease Epidemiology Collaboration (CKD-EPI) equation, which incorporates serum creatinine levels to assess renal function [6]. The ACR provides an additional measure of kidney health. According to current clinical guidelines, CKD is defined as eGFR of less than 60 mL/min/1.73 m², an ACR of greater than or equal to 30 mg/g, or both[7]. In the revised manuscript, we have explained that these two criteria define CKD. We hope this explanation addresses your concerns and clarifies the methodological rigor of our research.

Comment 5:

The authors may not have access to information about treatments of CKD, which could affect their models regarding eGFR, the stage of CKD, or kidney replacement therapy.

Response 5:

We agree that our analysis is limited by the lack of treatment data in the database. We have revised the Discussion section of the manuscript to include a discussion regarding this limitation and its impact on our results. We also emphasize that future research needs to incorporate treatment variables to provide a more comprehensive understanding of the CKD population. In keeping with your suggestions, we re-incorporated variables closely related to CKD that can be obtained from the database, such as eGFR, ACR, and C-reactive protein (CRP), in the analysis. We also made model adjustments and conducted subgroup analyses for the different categories of CKD. These revisions and analyses have been clearly outlined in the updated Methods and Results sections. Thank you again for your detailed review and valuable feedback, which greatly improved the quality of our manuscript.

References

1. Liu X, Wang Y, Lin C, et al (2025) Association between frailty index and mortality in depressed patients: results from NHANES 2005–2018. Sci Rep 15:3305. https://doi.org/10.1038/s41598-025-87691-4

2. Zhai C, Yin L, Shen J, et al (2024) Association of frailty with mortality in cancer survivors: results from NHANES 1999–2018. Sci Rep 14:1619. https://doi.org/10.1038/s41598-023-50019-1

3. Chang C-P, You C-H (2025) Serum selenium and reduced mortality in middle-aged and older adults with prefrailty or frailty: the mediating role of inflammatory status. Front Nutr 12:1560167. https://doi.org/10.3389/fnut.2025.1560167

4. Dai L, Schurgers LJ, Shiels PG, Stenvinkel P (2020) Early vascular ageing in chronic kidney disease: impact of inflammation, vitamin K, senescence and genomic damage. Nephrol Dial Transplant: Off Publ Eur Dial Transpl Assoc - Eur Ren Assoc 35:ii31–ii37. https://doi.org/10.1093/ndt/gfaa006

5. Li W, Chen S, Wan J, et al (2025) Comprehensive dietary antioxidant index and chronic kidney disease: mediating role of frailty and its impact on mortality outcomes in adults. Front Nutr 12:1679774. https://doi.org/10.3389/fnut.2025.1679774

6. Levey AS, Stevens LA, Schmid CH, et al (2009) A new equation to estimate glomerular filtration rate. Ann Intern Med 150:604–612. https://doi.org/10.7326/0003-4819-150-9-200905050-00006

7. Kidney Disease: Improving Global Outcomes (KDIGO) Glomerular Diseases Work Group (2021) KDIGO 2021 clinical practice guideline for the management of glomerular diseases. Kidney Int 100

---

## [Decision Letter · Decision Letter 1]

11 Jan 2026

How Frailty Index Impacts Death in Chronic Kidney Disease: A Retrospective Observational Investigation

PONE-D-25-41253R1

Dear Dr. Lu,

We’re pleased to inform you that your manuscript has been judged scientifically suitable for publication and will be formally accepted for publication once it meets all outstanding technical requirements.

Kind regards,

Ken Iseri

Academic Editor

PLOS One

Additional Editor Comments (optional):

Reviewers' comments:

Reviewer's Responses to Questions

**Comments to the Author**

Reviewer #1: All comments have been addressed

2. Is the manuscript technically sound, and do the data support the conclusions?

Reviewer #1: Yes

3. Has the statistical analysis been performed appropriately and rigorously?

Reviewer #1: Yes

4. Have the authors made all data underlying the findings in their manuscript fully available?

Reviewer #1: Yes

5. Is the manuscript presented in an intelligible fashion and written in standard English?

Reviewer #1: Yes

Reviewer #1: I. Assessment of Response to Review Comments

The authors have systematically addressed the core issues raised in the initial review, with most key suggestions effectively implemented. The overall quality of revisions is high.

II. Remaining Minor Issues

Inadequate clinical practical details: While the authors suggest integrating frailty management into CKD care, they have not specified clinically actionable intervention thresholds for the frailty index (e.g., whether FI>0.210 constitutes a high-risk threshold for intervention) or screening frequency (e.g., annual screening for CKD G3+ patients). It is recommended to briefly supplement these in the Discussion section to enhance clinical translatability.

III. Core Strengths of the Revised Manuscript

Significantly improved methodological rigor: Added CKD stage-specific subgroup analysis and mediation analysis, adjusted for key confounders such as ACR and CRP, and the statistical model is more aligned with the pathophysiological characteristics of CKD, enhancing the reliability of results.

Convincing conclusions: Using a large, nationally representative cohort (3,262 cases), the study clearly demonstrates a dose-response relationship between the frailty index and all-cause, cardiovascular, and cancer mortality. This association remains robust across subgroups of different CKD stages, sexes, and ages, ensuring strong generalizability.

Comprehensive and objective discussion of limitations: The authors candidly acknowledge data constraints (e.g., lack of treatment information, self-report bias) without avoiding methodological shortcomings, reflecting the scientific rigor of the study.

IV. Overall Review Conclusion

The authors have fully addressed the core issues raised in the initial review. The revised manuscript meets PLOS ONE’s scientific rigor requirements in methodology, statistical analysis, and result interpretation. The core conclusions are reliable and hold significant clinical and public health value. The remaining minor issues do not affect the core quality of the study, and no further major revisions are needed.

**Do you want your identity to be public for this peer review?** For information about this choice, including consent withdrawal, please see our Privacy Policy

Reviewer #1: No

---

## [Editor Report · Acceptance letter]

PONE-D-25-41253R1

PLOS One

Dear Dr. Lu,

I'm pleased to inform you that your manuscript has been deemed suitable for publication in PLOS One. Congratulations! Your manuscript is now being handed over to our production team.

Kind regards,

on behalf of

Dr. Ken Iseri

Academic Editor

PLOS One